# Radiative forcing due to shifting southern African fire regimes

Tom Eames<sup>1,2</sup>, Nick Schutgens<sup>1</sup>, Eleftherios Ioannidis<sup>1,\*</sup>, Ivar R. van der Velde<sup>1,3</sup>, Max J. van Gerrevink<sup>1</sup>, Roland Vernooij<sup>4</sup>, and Guido R. van der Werf<sup>4</sup>

**Correspondence:** Tom Eames (t.eames2@exeter.ac.uk)

Abstract. Landscape fires emit climate-influencing greenhouse gases and aerosols. The vast majority of landscape fire emissions originate from tropical savannas, especially in Africa. During the fire season climatic conditions change, and fires burning later consume drier vegetation and occur in drier weather conditions than earlier fires. Previous studies have shown that it is possible to reduce emissions of some greenhouse gases (CH<sub>4</sub> and N<sub>2</sub>O) by using 'prescribed' fires, i.e. deliberate burning in the early dry season. In this study we examine the climate effect of (deliberately) changing fire regimes beyond CH<sub>4</sub> and N<sub>2</sub>O, including aerosols and other short-lived species, CO<sub>2</sub>, and changes to surface albedo. We find that in general shifting burning earlier in a single fire season results in global negative climate forcing (cooling) of around -0.001 to -0.002 Wm<sup>-2</sup> (long-term) or -0.006 (short-term) Wm<sup>-2</sup>, compared to less than -0.0005 Wm<sup>-2</sup> if only considering CH<sub>4</sub> and N<sub>2</sub>O. Forcing from shifting burning later in contrast is negligible in the long term. CO<sub>2</sub> emissions reduction through emission factor changes and burned area reduction is the largest contributing factor, though especially in the short term albedo effects are also substantial. Shifting fire activity towards the late fire season generally produces a positive climate forcing (warming) of a smaller magnitude. We find too that some localities within our study area have a potentially disproportionately large impact on our results, such that the efficacy of any fire regime change with respect to climate forcing must be carefully considered on a local scale.

## 1 Introduction

Global emissions from landscape fires, colloquially referred to as 'forest' or 'wild' fires, total at least 2 Pg carbon annually (van der Werf et al., 2017). This is substantial when compared to total carbon emissions from fossil fuel, currently around 10 Pg carbon (Friedlingstein et al., 2023), though the vast majority of carbon emissions from biomass burning (BB) is re-sequestered provided there is post-fire vegetation re-growth (Landry and Matthews, 2016). Two thirds of global BB emissions come from the tropical savanna biome, of which most are from sub-saharan Africa (van der Werf et al., 2017), driven by a frequent burngrowth cycle typical of this region (Archibald et al., 2010).

Landscape fire emissions in this part of the tropical savanna consist mostly of greenhouse gases (GHGs) such as carbon dioxide

<sup>&</sup>lt;sup>1</sup>Department of Earth Sciences, Faculty of Science, Vrije Universiteit Amsterdam, Amsterdam, The Netherlands

<sup>&</sup>lt;sup>2</sup>Faculty of Environment, Science and Economy, University of Exeter, *Exeter, United Kingdom* 

<sup>&</sup>lt;sup>3</sup>SRON Space Research Organisation Netherlands, *Leiden, The Netherlands* 

<sup>&</sup>lt;sup>4</sup>Meteorology and Air Quality Group, Wageningen University & Research, Wageningen, The Netherlands

<sup>\*</sup>Now at Research & Development Satellite Observations, Royal Netherlands Meteorological Institute (KNMI), *De Bilt, The Netherlands* 

(CO<sub>2</sub>), methane (CH<sub>4</sub>) and nitrous oxide (N<sub>2</sub>O), short-lived gases such as carbon monoxide (CO) as well as other non-methane volatile organic compounds (NMVOCs) (Andreae, 2019; Urbanski, 2014; Yokelson et al., 2013). Aerosols emitted from BB are typically categorised into organic carbon (OC), black carbon (BC) and brown carbon (BrC) (Andreae, 2019; Bond et al., 2013; Feng et al., 2013) due to their differing optical properties; OC aerosols are generally light-scattering (Charlson et al., 1992), whereas BC and BrC are light-absorbing (Laskin et al., 2015).

Aerosol-radiation interactions (ARI; Carter et al., 2021; Canut et al., 1996), aerosol-cloud interactions (ACI; Tosca et al., 2014; Logan et al., 2024), climate warming effect from (pyrogenic) GHGs (IPCC, 2023; Etminan et al., 2016; Meinshausen et al., 2017), mean that BB emissions can affect the global climate in complex ways. Additionally, aerosol-climate interactions are less certain than those attributable to GHGs (Carslaw et al., 2010; Forster et al., 2021). Uncertainty in ARI is driven by several factors, including uncertainty in the underlying mechanisms, but also by a large spread in multi-model ensembles used to quantify it (Peace et al., 2020). ACI uncertainty is contained mostly in the uncertainty of the 'susceptibility' of cloud droplet number to aerosol load (Gryspeerdt et al., 2023).

GHGs drive long-term climate warming - CH<sub>4</sub> is the shortest lived with a lifetime of around a decade (Prather et al., 2012; Stevenson et al., 2020), N<sub>2</sub>O over 110 years (Chipperfield et al., 2014; Prather et al., 2015) and CO<sub>2</sub> can remain in the atmosphere from months to millennia depending on the removal process (Ciais et al., 2013). Aerosol lifetimes are far shorter, on the order of days to weeks (Kristiansen et al., 2016; Myhre et al., 2013). BC and BrC induce warming, contributing directly 70% and 20% respectively to aerosol-related absorption of sunlight, with the remaining 10% attributed largely to non-absorbing aerosols (e.g. sulfates) coating BC or BrC (Feng et al., 2013). OC by contrast has a net cooling influence, by direct light scattering but also via secondary processes such as assisting in cloud formation (Lu et al., 2015). The overall effect of pyrogenic aerosols is cooling (Tian et al., 2022; Xu et al., 2021) due to relatively higher quantities of OC in fire plumes than BC or BrC (Andreae, 2019; Bond et al., 2013).

The proportion of each of these climate-influencing species within a given fire plume depends on a number of factors including the type of vegetation being burned, the condition of that vegetation (e.g. moisture content), local weather at the time of the fire and longer-term weather conditions (Vernooij et al., 2023) as well as fire intensity (i.e. flaming vs smouldering fires; Laris et al., 2021). For example, wetter and/or coarser fuel combusts less efficiently and produces proportionally more CH<sub>4</sub>, N<sub>2</sub>O, CO and OC and less CO<sub>2</sub> than comparatively drier or finer fuel (Laris et al., 2021, 2023; Vernooij et al., 2022). In sub-saharan Africa fuel conditions are largely dictated by the timing of a particular fire. The fire season is concentrated in a 5-7 month dry season where little to no precipitation occurs (Archibald et al., 2019), such that fuel has several months to dry out and fires in the earlier part of the season burn wetter fuel. Fires in the early dry season can therefore emit a higher proportion of CH<sub>4</sub>, CO and/or aerosols when compared to similar fires later in the season. However, later on in the dry season the type of fuel burning may shift from grasses and other light material which cures (dries out) quickly to heavier material such as coarse woody debris and leaf litter which has fallen during the season, which also tend to emit a higher proportion of CH<sub>4</sub> and/or N<sub>2</sub>O

(Vernooij et al., 2023).

With this in mind, it is conceivable that altering burning patterns within a single fire season could by extension alter the amount and/or proportion of climate influencing pyrogenic species. Previous studies indicate that deliberately setting fires earlier in the dry season, a practice known as prescribed burning, can reduce burned area (BA) by creating a mosaic of smaller burned patches (Price et al., 2012; Russell-Smith et al., 2013; Laris, 2002). In this way, prescribed burning can also reduce 'net' GHG emissions (i.e. CH<sub>4</sub> and N<sub>2</sub>O, but not CO<sub>2</sub> due to the assumed re-uptake of this gas through photosynthesis) by reducing total emissions, despite the seasonal emission factor (EF) dynamics of these species (Russell-Smith et al., 2021; Lipsett-Moore et al., 2018). The effect of tropical savanna fires on climate via surface albedo changes has been the subject of some study (Dintwe et al., 2017; Jin and Roy, 2005), though not yet in the context of climate change mitigation. Aerosols in sub-Saharan Africa are substantially pyrogenic in origin (Andersson et al., 2020), but to our knowledge the influence of changing fire regimes or prescribed burning on aerosol emissions has not yet been studied.

70

The goal of this study is to take a comprehensive view of the climate influence of prescribed fire, and changes to burning patterns more generally, in southern Africa. We examine a series of burning scenarios and their influence on radiative forcing (RF) when compared to a pre-defined baseline burning scenario. We include long-term forcing from key GHGs (including CO<sub>2</sub>, often absent from the literature due to an assumed re-uptake in post-fire re-growth) via simplified parameterisations (Etminan et al., 2016; Moubarak et al., 2023), RF from aerosols and other short-lived climate forcers (SLCF) modeled using the Weather Research and Forecasting model with coupled chemistry (WRF-Chem, Grell et al., 2005), and RF from changes to surface albedo. In doing so our main aim is to offer perspective on emissions mitigation programmes involving savanna burning currently operating in e.g. Australia (CER Australia, 2015) and being expanded into the African continent (Russell-Smith et al., 2021), but we also consider scenarios with alternative burning patterns.

While the success of the Australian project is the motivating factor for this southern African study, it is also important to acknowledge that there are different considerations for the African continent. These considerations (including biodiversity goals and impacts on ecosystems and livelihoods) are detailed in Knowles et al. (2025), where the authors conclude that more evidence is required to show that a shift in fire timing will reduce greenhouse gas emissions. This work is an attempt rectify this, focusing on the climate impacts of such fire regime change to add to the discussion on whether such burning is desirable in the larger social, ecological and financial contexts of southern Africa.

We begin first with an introduction to the area of study, followed by an explanation of the burning scenarios, methodology and data we use to generate savanna emissions for a single fire year. We then describe in more detail the WRF-Chem model, including our experimental set-up, associated internal model choices and other data chosen to validate model runs for the chosen region. After this we present the methodology for albedo and longer-lived GHG forcing, and end the methods section with some of the ways we have mitigated uncertainty in this project. In the following sections we present first the results, a detailed discussion of these, and finally some concluding remarks.

#### 2 Methods

100

In the following section we describe the steps we took to quantify the influence of a change in fire patterns on global RF. We first establish the boundaries of the study area, then elaborate on how we constructed each burning scenario within this domain. Following this, we set out the methodology used to calculate RF from each of the three forcing components (SLCF, long-term GHG forcing and forcing due to surface albedo changes).

All RF figures we report are global and cumulative in time. RF from changes in GHG concentrations are calculated in the global context directly (see section 2.5), but we must scale forcing from SLCF and albedo with the fraction of Earth's surface covered by our domain, approximately 6%. For albedo effects this is a straightforward scaling, but for SLCF effects we must consider 'leakage' from e.g. aerosols from the WRF-Chem domain to the outside. We describe how we estimate this leakage in the SLCF section.

## 2.1 Area of study

This study defines 'southern Africa' as the region of the African continent south of the equator, excluding Madagascar, and we focus on fire patterns in the tropical savanna within this domain. Southern Africa is largely (sub)tropical (0°S to ~ 37°S), with a matching tropical climate. This involves two main seasons: a prolonged annual 'dry' season with little to no precipitation (April - October), and a 'wet' season during which the majority of the annual precipitation occurs (November - March). Following the biome classification system proposed by Olson et al. (2001), southern Africa can be split into several biomes (Figure 1). Dominant across the region is the tropical and subtropical grasslands, savannas and shrublands biome, hereafter simply 'savanna'. This biome is typically characterised by open woodland containing a mixture of trees and grasses (Beerling and Osborne, 2006), and has a strong relationship with fire; southern African savannas are frequently burnt, on average every five years and in some areas annually (Archibald et al., 2010; Giglio et al., 2018).

Our domain is shown in Figure 1, and we concern ourselves primarily with fire patterns in the savannas south of the equator. This area contains several sparsely populated and protected areas, where fires tend to be more spatially extensive and occur later in the dry season (Eames et al., 2023; Archibald et al., 2010). In this paper we only address changes to global RF as a result of shifts in southern African fire patterns, and all numbers presented should be interpreted in this context. We do not concern ourselves with the absolute forcing as a result of tropical fires as a whole, but rather the difference relative to a pre-defined baseline scenario.

#### 120 2.2 Scenarios

We created six scenarios in total. We examine effects of shifting fire activity to either earlier or later periods in the fire season than a baseline 'average' fire year, keeping total BA constant. Additional scenarios were based on the same intra-seasonal shifts, but including additional changes in total BA. Full details are given in the following subsection.

**Figure 1.** Map of the study area which makes up the entire WRF-Chem domain. The WRF-Chem domain includes the 'ground' domain where emissions inputs are varied, corresponding to the yellow 'Tropical and Subtropical Grasslands, Savannas and Shrublands' region south of the equator. Also displayed for the region is the World Wildlife Fund terrestrial ecoregion classification (Olson et al., 2001). We include fixed emissions for areas north of the equator (red dotted line) and Madagascar, though these are unchanging across all scenarios.

Our analysis is based on these scenarios in a single fire season, and longer-term results represent the cumulative effect of changing fire dynamics in that single season. We do not account for future fire seasons in this experimental set-up.

#### 2.2.1 Scenario design & burned area

We designed each scenario using BA data developed for use in the newest Global Fire Emissions Database (GFED5) from Chen et al. (2023). We defined the baseline scenario as the mean monthly BA over the period 2003-2020, in order to best represent an average fire year.

We based our other scenarios on idealised gaussian distributions with BA peaks in different parts of the dry season: the early

dry season (EDS) distribution peaks in mid June, while the late dry season (LDS) distribution peaks in mid August (Figure 2a). The relative widths of the EDS and LDS distributions reflect fire weather conditions throughout any given season: fire weather in the EDS is generally less conducive to spatially extensive burning than in the LDS (Williams et al., 2003; Perry et al., 2019). The opposite is true in the LDS. This means that large areas can burn quickly in the LDS, but not in the EDS. A realistic shift to more EDS burning is therefore spread more evenly across the season than an LDS shift.

These distributions defined how BA could be shifted from the baseline (mean) scenario to a new simulated fire regime, in combination with a grid cell specific transition date from EDS to LDS. This transition date was determined by the proportion of night-time fires in total active fire detections (see Eames et al., 2023 for details on this). To determine how much BA can be shifted, we take the percentage of that BA which falls on the opposite side of the transition date (i.e. after for EDS scenarios and before for LDS scenarios), and redistribute this over the relevant gaussian. A summary of all six scenarios is given in Table 1.

**Table 1.** Summary table of the scenarios used in this paper. The degree of change is dependent on the proportion of burning already taking place before the cut-off date (EDS or LDS) in the grid cell, up to a maximum of 15% for the scenarios where there is a change in total BA.

| Scenario        | Change in BA | Summary                                             |
|-----------------|--------------|-----------------------------------------------------|
| Mean (baseline) | _            | The monthly mean of all BA from 2003-2020 from      |
|                 |              | GFED5 BA data                                       |
| EDS             | None         | LDS BA shifted towards the EDS                      |
| LDS             | None         | EDS BA shifted towards the LDS                      |
| EDS reduction   | Up to -15%   | LDS BA shifted towards the EDS and reduced by up to |
|                 |              | 15%                                                 |
| LDS increased   | Up to +15%   | EDS BA shifted towards the LDS and increased by up  |
|                 |              | to 15%                                              |
| EDS suppressed  | Up to -15%   | EDS BA reduced by up to 15%                         |

To illustrate the process of shifting BA towards the EDS, we use a grid cell located in central Zambia as an example (Figure 2b). To shift BA earlier in this grid cell, there must be burning in the LDS available to shift. 37% of the BA in this particular grid cell occurs after the transition date (based on Eames et al., 2023). We thus create an EDS gaussian containing 37% of the BA, and reduce the total BA by that 37% (EDS remainder). By summing these two BA distributions a new BA distribution is created, with the same total BA but shifted towards the EDS. This is expressed in equation 1:

$$BA_{scenario,i} = \beta_i \times BA_{gaussian,i} + (1 - \beta_i) \times BA_i \tag{1}$$

where  $\beta_{s,i}$  is the fraction of BA in the LDS or EDS (depending in the scenario) in the  $i^{th}$  grid cell. This equation represents scenarios where there is no change in total BA.

**Figure 2.** Conceptual representation of scenarios (a) of BA in the EDS (blue) and LDS (red) along with mean BA from 2003-2020 (black). All areas under the curve are identical, i.e. total BA does not change A value of 1 on the y axis corresponds to the maximum monthly BA in the Mean scenario. (b) shows the process of shifting burned towards the EDS in a grid cell in central Zambia (-13.6°N, 29.1°E). For this grid cell only 37% of total BA is in the LDS and can be shifted to the EDS gaussian. The remainder is not shifted, but reduced by 37%. The BA distribution in this grid cell in the EDS scenario is given by the sum of the gaussian and reduced components. In this particular example, the total BA also remains consistent.

For scenarios with a change in BA, each grid cell is scaled with the late season burn index (LSBI) for that grid cell. The LSBI is a quantity related to both how much BA there is in a particular grid cell, and what fraction of that occurs in the LDS (Eames et al., 2023). An LSBI close to 1 indicates a relatively large BA very late in the season, and close to 0 indicates small BA much earlier. Areas with a higher LSBI see a greater reduction of BA due to the shift to EDS fires. The three scenarios with change to total BA are:

- EDS reduction: in some areas where the introduction of more EDS fire has been studied, total BA reductions can be around 15% (Edwards et al., 2021; Price et al., 2012). We therefore also include a scenario where total BA is reduced as a consequence of a shift towards EDS burning, whereby the level of reduction per grid cell is scaled by the LSBI up to a maximum of 15%.
- LDS increase: prescribed EDS burning is already practiced in many parts of southern Africa. Were this activity to be stopped, in line with the previous scenario we may imagine a shift of fire activity in the direction of the LDS. Additionally, if we assume that EDS burning may reduce BA, we may also assume that BA may increase in an LDS-shifted scenario by a similar amount. In this scenario fire-activity is LDS-shifted and BA increases by up to 15% in certain areas. In this instance, areas with a high LSBI see a small BA increase, and vice versa.
- EDS suppression: we imagine a scenario in which resources typically dedicated to EDS burning are instead channeled towards the suppression of fires. Suppression is more achievable in the EDS when fires are cooler and less extensive, and as such the reduction in BA occurs predominantly in the EDS. The result is that fire activity is effectively LDS-shifted (less burning in EDS, relatively unchanged burning in LDS). BA reduction is achieved in areas with high EDS burning, again up to 15%.

It is important to note that we use the terms 'EDS' and 'LDS' to indicate relative change, and in our scenarios these should not be understood to refer to specific times of the year. It is possible for example that an 'EDS shift' for a grid cell with a very late-skewed fire season may still result in burning being shifted to e.g. August. The spatial distribution of BA changes in the three scenarios itemised above is shown in Figures S2, S3 &S4 in the supplementary material.

#### 175 2.2.2 Scenario emissions




Pyrogenic emissions from tropical savannas in each scenario were calculated following Seiler and Crutzen (1980):

$$Emissions_{species} = FL \times BA \times CC \times EF_{species}$$
 (2)

where FL is total available fuel load, BA is the scenario-specific burned area as described in the previous subsection, CC is the fraction of the fuel combusted (combustion completeness) and EF is the emission factor of the emitted species.

# 180 Fuel consumption

FL and CC input for surface fuel classes (grass, litter, coarse woody debris) and shrubs savanna emissions data was created by building a Sentinel-2 scale fuel map from in-situ measurements. We performed Unmanned Aerial Vehicle (UAV) surveys over plots where FL measurements were taken for the different fuel classes. By combining these surveys with meteorological data we created a set of surface fuel map 'samples' (Eames et al., 2021). Using these samples, we scaled the FL map up to Sentinel-2 tiles by building a machine learning model, and these tiles could then be stitched together to create regional and continental scale fuel maps (Eames et al., 2025).

Tree biomass data was generated using high resolution synthetic aperture radar (SAR) data (Bouvet et al., 2018). Monthly FL and CC maps were generated for the entire study region for the year 2019 on a 500m pixel scale. FL varies somewhat over the fire season, in particular with the accumulation of nitrogen-rich leaf litter and more woody fuel as a result of litterfall and tree/shrub mortality in the dry season. CC generally scales with moisture content and will increase from a minimum at the start of the dry season to a maximum after sufficient time without rainfall, though the drying rate depends on the type of fuel (for example, grasses dry out faster than shrubs and thus burn more completely earlier in the season). More details can be found in Eames et al. (2025).

To match the FL maps to the GFED5 BA resolution, we averaged FL across all 500m pixels contained within a single  $0.25^{\circ} \times 0.25^{\circ}$  GFED5 grid cell.

#### BA




BA data was taken from the GFED5 dataset (Chen et al., 2023). This dataset is derived from the MODerate Resolution Imaging Spectroradiometer (MODIS) but includes 'small fires' (in general less than 100ha) on a statistical basis. These small fires are crucial to accurate BA estimates in all sub-Saharan Africa, as they could account for up to half of total BA (Ramo et al., 2021).

#### 200 Emission factors

EFs were generated using a combination of measurements from a UAV-mounted smoke sampling system (Vernooij et al., 2022), and other existing EF measurements globally (e.g. Akagi et al., 2011; Andreae, 2019). For tropical savannas this database was coupled with climate reanalysis data and vegetation data from MODIS products to generate an EF product which is both temporally and spatially dynamic, and is able to distinguish between areas of high grass density or more closed canopy cover (Vernooij et al., 2023). To match the BA data, we used the aggregated monthly dataset on a  $0.25^{\circ} \times 0.25^{\circ}$  resolution (Vernooij, 2023). We accounted for three primary pyrogenic GHG EFs:  $CO_2$ ,  $CH_4$  and  $N_2O$ . We also included  $CO_3$  which despite not being a GHG itself impacts global OH and ozone ( $O_3$ ) concentrations (Lelieveld et al., 2016), and ultimately contributes to total  $CO_2$  concentrations via oxidation (Crutzen, 1973).

CO<sub>2</sub> EFs increase later into the fire season as the fuel gets drier, and the opposite is usually true for CO and CH<sub>4</sub>. For N<sub>2</sub>O the type of fuel itself is a key determining factor (Vernooij et al., 2023). OC aerosol EFs are linearly related to the modified

combustion efficiency (MCE), itself calculated from the ratio of  $CO_2$  to  $CO + CO_2$  emissions (Vernooij et al., 2022). More efficient burning is less conducive to OC production than inefficient combustion, and combustion efficiency generally increases as the dry season progresses (Vernooij et al., 2021, 2023), though can also be affected by fuel composition (Ward et al., 1996). We calculated  $EF_{OC}$  from the following equation (Vernooij et al., 2022):

$$EF_{OC} = -116 \times MCE + 116$$
 (3)

BC aerosol EFs are not linearly related to the MCE (Andreae et al., 1998; Vernooij et al., 2022), and accounting for the  $EF_{BC}$  variability is not straightforward. At the time of writing the authors are unaware of good data regarding seasonal changes in  $EF_{BC}$  in southern African savanna fires. We therefore retained the fixed  $EF_{BC}$  value for tropical savannas as in van der Werf et al. (2017). All other pyrogenic species are based on similar fixed values for the tropical savanna biome from Akagi et al. (2011), also consistent with GFED4.1s.

#### **2.3 SLCF**




We define SLCFs as those forcing agents directly and indirectly generated by pyrogenic emissions and limited in lifetime on the order of weeks to months, such that the effects do not extend far beyond the fire season itself. This includes forcings from aerosol effects, and some short-lived gas species. Aerosols in particular can influence the climate in many and sometimes opposing ways by scattering (cooling), enhancing cloud production (mostly cooling), or absorbing incoming solar radiation (warming). The 'aerosol effect' should be understood hereafter to refer to the sum total of all of these processes.

The total forcing from these species is calculated on-line within each WRF-Chem scenario simulation, itself then compared to a baseline model run to calculate the total RF for that particular scenario in a single fire year. Long-lived GHGs also affect forcing within the WRF-Chem simulation, and to avoid double-counting we subtract this RF component from the total WRF-Chem RF to obtain RF solely from short-lived species (forcing from long-lived GHGs is separate to the WRF-Chem simulation and outlined in section 2.5). The contribution from SLCFs on the longer term represents the cumulative RF from the WRF-Chem simulation period, and the contribution from that specific fire season over the following period (which is zero).

# 2.3.1 WRF-Chem model

The Weather Research and Forecasting model with coupled chemistry (WRF-Chem, Grell et al., 2005) is a regional mesoscale atmospheric model capable of simulating aerosol concentrations, optical properties and the effect thereof on cloud formation, and also RF more broadly (e.g. Fast et al., 2006; Ahmadov et al., 2012; Ntelekos et al., 2009; Ha, 2022).

In our scenario runs with WRF-Chem, gas-phase chemistry was parameterised using an updated version of the Regional Atmospheric Chemistry Mechanism (RACM, Stockwell et al., 1997), allowing for inclusion of an extensive list of NMVOCs as well as key well-mixed GHGs and their precursors. The Modal Aerosol Dynamics Model for Europe (MADE, Ackermann et al., 1998) was used to parameterise inorganic aerosols, and organic aerosols were treated with the volatility basis set (VBS) model with simplified aqueous chemistry (Ahmadov et al., 2012; Tuccella et al., 2015) allowing for the on-line inclusion of both the direct and indirect effect of aerosols on RF. Aerosol optical properties were calculated on-line using Mie theory following

Tuccella et al. (2015). We used a single domain with grid resolution set to 30km×30km.

We ran the WRF-Chem model for a single seven-month period within each scenario using meteorology, initial and boundary conditions from the year 2019. These periods began on April  $1^{st}$  and ended on November  $1^{st}$ , including a two week spin-up period (1-14 April). This time period encompasses the vast majority of the southern African savanna fire season, including both very early and very late dry season fire activity (Archibald et al., 2010). The model was run separately for each burning scenario as in Section 2.2, as well as a validation run (see Section 2.3.4). RF from WRF-Chem was calculated from the difference in the total all-atmosphere up- and down-welling radiative flux (long-wave and short-wave) between each scenario and the baseline.

We based our validation run on observational data for the 2019 fire season, as much of the field data was collected in this period. Input data for all runs such as meteorology and anthropogenic emissions also corresponded to the 2019 fire season, and were kept constant in all scenarios. Input data sources are described in the following subsection.

#### 2.3.2 WRF-Chem input data

# Meteorology input






We sourced the 2019 meteorological data from the National Center for Environmental Prediction's (NCEP) final (FNL) Operational Global Analysis (NCEP, 2000), available on a 1°×1° grid for the study period in 2019. WRF can resolve its own meteorology from initial and boundary conditions, but in order to keep our simulations close to the FNL reanalysis data we nudged the WRF meteorology with FNL data every six hours. Nudging improves estimations of temperature and moisture variables only if not applied too strongly (Alexandru et al., 2009), so appropriate nudging coefficients were calculated (x and y wavenumbers of 7 and 6 respectively) for our model resolution and domain size using methods described in Spero et al. (2018). This approach ensures that WRF internal meteorology remains as close to observed conditions as possible. We confirmed this with two shorter test runs with identical input data, in which differences between meteorology were negligible. Using the same reanalysis data and nudging in other model runs also ensures consistency between scenarios, though any feedback from the fires on the meteorology are outside the scope of this study.

#### Anthropogenic and biogenic emissions

Biogenic and anthropogenic emissions (excluding BB emissions) were sourced from the Copernicus Atmosphere Monitoring Service (CAMS) version 3.0 and 5.3 respectively (Granier et al., 2019; Soulie et al., 2024). These emissions inventories are based on the EDGAR repository up to 2018 (Crippa et al., 2021) and extrapolated for later years based on emissions trends from O'Rourke et al. (2021). These are available globally on a  $0.1^{\circ} \times 0.1^{\circ}$  grid for a range of key GHGs, NMVOCs and aerosol species. We interpolated these data onto the WRF domain grid using bilinear interpolation, ensuring minimal (

## 2.3.3 Domain leakage

Transport of aerosols and other short-lived species out of the domain represents a 'loss' of possible RF in each scenario. These species will affect global RF, but are not captured by the WRF-Chem simulations as this forcing occurs outside of the domain boundary. To estimate the size of this deficit, we used aerosol optical depth (AOD) as an indicator variable. For each scenario we fitted a gaussian curve to the tail ends of both zonal and meridionally-averaged AOD to roughly follow the decay of AOD as it passes outside of the domain. This enabled us to estimate a fractional 'loss' of RF-influencing species due to transport out of the domain (Figure 3). The total SLCF forcing component (RF<sub>SLCF</sub>) from within the domain was adjusted by the percentage loss calculated. By using the ratio of the model domain area to total global surface area we may scale WRF-Chem regional forcing up to a global RF estimate, accounting for outflow across the model boundary.

Aerosols are only one component of forcing within the WRF-Chem scenarios, so this method does not directly represent RF leakage. However, we are largely estimating a transport effect, one which will affect other SLCF species in a similar manner.

**Figure 3.** Zonal and meridional domain leakage in the EDS scenario. Transport patterns in the simulations mean that leakage occurs more or less exclusively at the western and northern boundaries.

#### 2.3.4 WRF-Chem validation

To assess the performance of WRF-Chem as pertains to biomass burning emissions, we used a combination of satellite-based CO and both satellite and ground-based aerosol observations for the year 2019.

#### TROPOMI

The TROPOspheric Monitoring Instrument (TROPOMI) is mounted on the Sentinel 5 Precursor (5P) satellite, itself in a sunsynchronous orbit such that it passes overhead at 13.30 local time. Measurements are taken on a nominal 7km×7km grid of various (climate-affecting) species including O<sub>3</sub>, CH<sub>4</sub>, and CO. CO in (southern) Africa is substantially pyrogenic in origin (van der Velde et al., 2024), which along with a relatively short lifetime of about 1 month (Khalil and Rasmussen, 1990) makes it an appropriate species to base this part of our validation on. CO retrievals from TROPOMI are derived from short-wave infrared radiance via the Shortwave Infrared Carbon Monoxide Retrieval (SICOR) algorithm (Vidot et al., 2012; Landgraf et al., 2016).

We compared atmospheric column CO from high-quality TROPOMI observations, filtered to be cloud free or limited to containing low-level cloud up to 5km, to WRF column CO. Column retrievals from WRF were adjusted for layer sensitivity changes within the TROPOMI product using the column averaging kernel (Borsdorff et al., 2014). The mean WRF CO column between 12:00 and 15:00 was used to most closely align with the TROPOMI overpass times. We aggregated both WRF and TROPOMI datasets to a  $0.5^{\circ} \times 0.5^{\circ}$  grid for ease of comparison.

#### 320 AERONET

The AErosol RObotic NETwork (AERONET) is a network of over 600 sun photometers placed in various locations around the globe. These passive sensors measure column AOD at a number of wavelengths using the Beer-Lambert-Bouguer extinction law (Holben et al., 1998) at high temporal resolution. Raw data from these stations is screened for anomalies, clouds and other possible issues which may affect the data quality, and is available for download on the AERONET website (https://aeronet.gsfc.nasa.gov/new web/aerosols.html, accessed 2024-06-16).

We used Level 2.0 AOD data from the version 3 AERONET database (Giles et al., 2019) from all stations within our domain which have data available for the 2019 southern hemisphere fire season (a total of 22 stations) AERONET data does not cover the exact wavelengths on which WRF outputs AOD (in this case 550nm), but as wavelength and AOD are approximately linearly related on a logarithmic scale (Tan et al., 2016) it is straightforward to apply a correction for this. In this case, we used observations at 500nm and 675nm to calculate AOD at 550nm.

We compared this 550nm AOD with the WRF-Chem column AOD at grid cells spatially and temporally co-located with valid AERONET data. WRF column AOD was calculated by summing the extinction coefficient at each vertical layer ( $b_{ext}$ ) multiplied by the vertical thickness of that layer ( $\delta z$ ):

$$AOD_{550,WRF} = \sum_{n=1}^{N} b_{ext,n,550} \, \delta z_n \tag{4}$$

where N is the total number of model layers. We then calculated an average daily AERONET AOD over the study domain (Figure 1), and an analogous average WRF AOD using output which corresponds only to locations and times where AERONET stations had valid data.

# **MODIS AOD**

MODIS-derived AOD from MCD19A2, or the Multi-Angle Implementation of Atmospheric Correction (MAIAC, Lyapustin and Wang, 2022) at 550nm is available on a daily basis on a 1km×1km grid. We chose this AOD retrieval algorithm over other algorithms (Dark Target or Deep Blue) as it has been shown to perform better over vegetated areas and, in particular, for smoke AOD (Mhawish et al., 2019). Measurements are taken of blue-band (470nm) AOD and converted to 550nm using the spectral properties of a regional aerosol model (Lyapustin and Wang, 2022). We aggregate this output to the WRF domain grid. The WRF data is then spatio-temporally collocated with the MODIS data to validate domain-scale AOD.

#### 345 **2.4** Changes in surface albedo

The surface albedo of the savanna is changed by fire in the short to medium term - in most cases the surface darkens as a result of charcoal and ash deposited from the smoke plume (Smith et al., 2005; Jin and Roy, 2005). The outcome of this is generally a reduction in the surface albedo, in particular in the near-infrared portion of the spectrum (Giglio et al., 2018; Roy et al., 2005). A decreasing surface albedo leads to less incoming short-wave solar radiation being reflected from the surface, resulting in an increase in RF (López-Saldaña et al., 2015).

Dintwe et al. (2017) determined that in southern hemisphere Africa, average regional RF over the fire season due to fire-induced albedo changes is around  $\pm 0.33 \, \mathrm{W}m^{-2}$ . They use the ratio of total burned area to global land surface area to scale this regional forcing to a global one. Doing the same using the total BA in the Mean (baseline) scenario, we arrive at a global forcing of  $\pm 0.046 \, \mathrm{W}m^{-2}$ . This influence is short-lived as vegetation recovers relatively quickly. These short-term albedo changes diminish by the start of the next wet season (assumed to be October) due to the onset of persistent cloud cover, or when vegetation recovers sufficiently after approximately three months on average (Dintwe et al., 2017). We used a simple parameterisation scaled by fire activity (BA) on a monthly basis to account for RF due to surface albedo changes in each scenario. When a pixel burns earlier than the co-located pixel in the baseline scenario, this provides more land surface warming due to earlier darkening (via ash and charcoal deposition) resulting in positive RF. For pixels that burn later, the opposite is true. We calculate this RF component by using a monthly cumulative sum of BA per scenario, and we maintain the assumption that savanna vegetation recovers within about 3 months as in Dintwe et al. (2017):

$$DA_{m} = \sum_{i=1}^{m} BA_{m} - \sum_{m=1}^{m-3} BA_{m} \quad if \ m < 9$$

$$= 0 \qquad if \ m \ge 9$$
(5)

where DA is darkened area, i.e. the total area currently contributing to surface albedo related forcing in any given month m. DA is then used to scale the total RF per month:

865 
$$RF_{\alpha,m,s} = 0.046 \times \left(\frac{DA_{m,s} - DA_{m,base}}{\sum_{i=1}^{12} DA_{i,base}}\right)$$
 (6)

where  $RF_{\alpha}$  is the albedo-induced RF per month m in scenario s.

It is important to note that this approach is effectively a per-month scaling of an annual average  $RF_{\alpha}$ . It does not take into account local cloud cover, smoke plume presence, or relative change in surface albedo from e.g. brighter or less densely vegetated patches compared to more dense cover.

#### 370 2.5 Long-term forcing from well-mixed GHGs

Well-mixed GHGs emitted from fires (such as  $CH_4$  and  $N_2O$ ) are much longer lived than surface albedo changes or aerosols (IPCC, 2023). These long lifetimes mean that GHG species exert RF for much longer periods than surface albedo changes or forcing from SLCFs.

To include long-term GHG forcing in our analysis of possible fire regime changes, we adapted the methods outlined in Moubarak et al. (2023) for the savanna biome, using emissions from the different fire scenarios as described in section 2.3.2. For ambient GHG concentrations, we assumed the Earth to be following a moderate climate mitigation trajectory as in the 'middle of the road' Shared Socio-economic Pathway (SSP) 2-4.5 (IPCC, 2023; Meinshausen et al., 2020) in our RF calculation. We include relevant long-lived pyrogenic species: CH<sub>4</sub>, N2O, and NMVOCs.

 $CO_2$  from non-deforestation vegetation fires may be viewed as having a shorter 'lifetime' than  $CO_2$  from fossil fuel sources due to post-fire regrowth of vegetation (Landry and Matthews, 2016), and in previous savanna fire emissions abatement studies it has not been included as a net contributor to RF (Lipsett-Moore et al., 2018; Russell-Smith et al., 2009; Russell-Smith et al., 2021). This is perhaps valid for substantially longer-term projections, but on a relatively shorter time scale this  $CO_2$  enters the atmosphere and can contribute to global RF. Positive  $CO_2$  anomalies are indeed observed over southern Africa during peak fire months (Hakkarainen et al., 2019). We therefore include  $CO_2$  in our forcing estimates, but tune the lifetime such that almost all pyrogenic  $CO_2$  has been re-sequestered within the time frame of 1 year. We must also account for potential changes to respiration of  $CO_2$  due to shifting fire patterns (i.e. if fuel does not burn in a scenario, it can still be respired as  $CO_2$ ). Details of how we quantify this are presented in section 2.6, but we effectively calculate two extremes: one with no change in respiration, and one with an estimated maximum potential change in respiration. The reported  $RF_{CO_2}$  is the mean of these two extremes, and the extremes themselves represent a large part of the  $RF_{CO_2}$  uncertainty.

Using the projected concentration increase as a result of the total emissions of each species in a fire season, we calculate changes in RF after the season has ended using the simplified expressions described in Etminan et al. (2016). We exclude CO from this portion of the analysis, as it has a short lifetime of around 1-3 months (Zheng et al., 2019) and (as with aerosols) RF effects thereof are captured within the WRF-chem scenario run. We also exclude O<sub>3</sub> as it was found to have negligible impact after the fire season (Moubarak et al., 2023). For the remaining relevant species, we calculated the annual RF in the 20 years following the fire season.

#### 2.6 Uncertainty mitigation




For uncertainties in  $RF_{\alpha}$  and  $RF_{GHG}$  excluding  $CO_2$ , we adopt the uncertainties reported in the respective publications; around 96% for  $RF_{\alpha}$  (Dintwe et al., 2017) and 2.7% for  $RF_{GHG}$  (Etminan et al., 2016). We also adopt the 3.6% uncertainty in  $RF_{CO_2}$  reported by Etminan et al. (2016), but add this to the additional uncertainty generated by respiration effects (see below). For uncertainty in  $RF_{SLCF}$ , we quantify the internal variability component of our WRF-Chem model runs.

# Internal model variability

A complex model such as WRF-Chem has many potential sources of uncertainty. Uncertainties for RF stemming from emissions input uncertainty and errors in internal model processes and parameterisation (cloud formation processes, aerosol size distribution, vertical transport speeds etc) manifest as biases (Zhong et al., 2023; Lohmann and Ferrachat, 2010). This should not affect differences between two model runs with the same biases, as in our experimental set-up. The same is true of differing hardware options on which it is possible to run WRF-Chem (Li et al., 2016). These types of uncertainty should cancel out in the final RF calculation.

Internal model variability (IMV) within WRF-Chem may affect the RF uncertainty, however. We cannot neglect IMV in the same way as the model or input errors, as this constitutes a more chaotic element within WRF-Chem (Bassett et al., 2020; Laux et al., 2017). We tested IMV in our setup by performing an additional simulation with a delayed start date, and compared

average RF between the two runs to calculate an internal variability factor,  $f_{IMV}$ :

$$f_{IMV} = \frac{\left| \overline{RF}(t_0, s) - \overline{RF}(t_0 + dt, s) \right|}{\overline{RF}(t_0, s)}$$
 (7)

where  $\overline{\text{RF}}$  is the mean RF for the scenario s starting at time  $t_0$ . In our test case,  $t_0$  was April 1<sup>st</sup> and dt was one week. We calculated  $f_{IMV}$  to be around 1%, small compared to that found in e.g. Bassett et al. (2020), though their study directly compared single runs whereas we are comparing the differences between two runs, a fact which seems to substantially reduce uncertainty from the WRF simulations in general.

#### 420 Respiration




We account for the photosynthetic re-sequestration of  $CO_2$  from vegetation re-growth by tuning the atmospheric lifetime of this species, but changes in fire patterns also have the potential to affect heterotrophic respiration patterns.  $CO_2$  emissions avoided by reducing BA in a given scenario may be compensated by the respiration of excess unburned organic material. There are potentially substantial differences in wet season respiration depending on whether an area has burned, and what type of vegetation is present. During the wet season, an area unburned in the preceding fire season may respire up to four times more  $CO_2$  than a similar area affected by fire, though the respiration rate declines exponentially with the loss of biomass (Richards et al., 2012). We also assume that all unburned areas contain substantial amounts of leaf litter, an unrealistic assumption but one which gives us the largest change in respiration rate, from 'dry season burned' respiration rate to 'wet season unburned'. For a simple expression, we assume that the respiration rate halves over the course of half of the wet season. Additionally, we also include ash deposition, which encourages a post-fire  $CO_2$  respiration pulse (Sánchez-García et al., 2021).

Seasonal respiration change in a given scenario is calculated as

$$\Delta R_{total} = \left(\sum_{d=1}^{181} r_s \ e^{-\tau/d} - p_s\right) \times \Delta B A_s \tag{8}$$

where  $r_s$  is the average respiration rate of 22 g CO $_2$  m $^{-2}$  day $^{-1}$  (Richards et al., 2012), d is the day over the course of the wet season, the exponential term ( $\tau=60$ ) represents the reduction in respiration as material is consumed,  $p_s$  is the average cumulative post-fire respiration pulse of 40 gCO $_2$  m $^{-2}$  burned (Sánchez-García et al., 2021), and  $\Delta BA_{scenario}$  is the difference in total BA between the scenario and the baseline.  $\Delta R_{total}$  can take both positive and negative values, depending on the sign of  $\Delta BA_{scenario}$ . RF $_{CO_2}$  can then be calculated as the average of RF only due to CO $_2$  changes  $RF(Emissions_{CO_2})$  and RF including respiration  $RF(Emissions_{CO_2} + \Delta R_{total})$ . These two 'versions' of RF $_{CO_2}$ , in combination with the 3.6% uncertainty from Etminan et al. (2016), represent the upper and lower bounds of RF due to CO $_2$ .

This parameterisation of respiration effects relies on there being a change in BA in a scenario - that is, we assume that respiration does not change based on *when* a patch of land burns, only *if* it burns. There may well be intra-seasonal changes in respiration from timing of fire and we do not account for these in this study. However, Richards et al. (2012) suggest that this

**Figure 4.** Temporally-averaged model column carbon monoxide (XCO) from (a) TROPOMI and (b) WRF-Chem from mid-April to the end of October.

effect is minimal, especially given that heterotrophic respiration is substantially lower in the dry (fire) season compared to the wet season (Fan et al., 2015).

#### 3 Results

#### 3.1 Model validation

# $\mathbf{CO}$


Model column CO (XCO) corresponds reasonably well with TROPOMI measured XCO. Both show peak mean XCO concentrations above western D.R.C/Eastern Congo and Northern Angola of around 200 ppb (Figure 4). The temporal correlation coefficient ( $r^2$ , excluding model spin-up) for spatially averaged WRF and TROPOMI XCO is 0.76 (RMSE = 23.4, see Figure 5). Over the full domain WRF generally underestimates XCO (though peak WRF XCO is higher than that of TROPOMI), especially in the mid-late dry season (June - August) where fire activity is at its highest (Figure 5). This underestimate could be due to fires being missed in the dataset, such as smaller fires or sub-canopy fires which can remain undetected by BA products (Ramo et al., 2021). Smouldering fires which produce proportionally higher CO emissions (Vernooij et al., 2023; Johnston et al., 2018) are especially prone to remaining undetected as they are relatively cooler, and far less spatially extensive. Additionally, van der Velde et al. (2024) showed that part of the CO column over Africa originates from fires in Indonesia and the Amazon. Underestimation of the inflow from this CO could be why XCO is underestimated by WRF in the southern and eastern parts of the domain.

**Figure 5.** Spatially averaged XCO for both TROPOMI and WRF output over the study area. The blue highlighted area is the model spin-up period.

Another possible explanation for the XCO peak values in WRF being higher than the observations is a reduced lateral transport in the model, allowing for a CO build up in one place (notable in the northwest). This may also partially cause underestimations elsewhere in the domain - although it is possible that regional under- or overestimates in the emissions input variables (BA, FL, CC or EF) are also playing a role.

#### **AOD**

AOD differences between satellite (MODIS) observations and WRF output are similar to those between TROPOMI and WRF XCO. The spatial agreement between observational data and model output is generally strong, with both showing a peak in AOD in similar locations. However, the MODIS data show aerosols as being more widely distributed across the domain, whereas WRF indicates aerosols are spatially more densely packed around the peak AOD area (in red in Figure 6). Notably, some drier regions have a higher observed AOD than found in WRF, such as the Etosha pan (Namibia) and the eastern parts of Kenya and Ethiopia, indicating that perhaps aerosols originating from dust or sand over more arid landscapes are underestimated. Cities such as Gaborone (Botswana) and Antananarivo (Madagascar) are also observed as stronger aerosol sources than simulated in WRF. This too is indicative of an underestimate of anthropogenic emissions for these locations.

**Figure 6.** 550nm AOD for the study area from (a) MODIS MCD19A2 product and (b) WRF-Chem from mid-April to the end of October. The WRF-Chem output has been temporally and spatially co-located with the MCD19A2 product to best represent the comparison between the two. White areas indicate regions of no data in the MCD19A2 product.

In the early part of the fire season WRF AOD matches well with both MODIS and AERONET data (Figure 7). In June WRF tends to over-predict AOD (sometimes by as much as 50%), and from August onwards WRF can under-predict AOD to a somewhat lesser degree (about 40% in extreme cases). In particular for the AERONET data, this is an average across multiple stations which do not necessarily all overlap temporally. Large discrepancies between observed and modelled AOD could thus be driven by a single station. Another possible (partial) explanation may be the use of static EFs for aerosol species other than OC, limiting the accurate representation of temporal trends in aerosol emissions. Early in the season, mostly light surface fuels are being burnt, resulting in less smouldering and lower aerosol production. As the season advances, this situation is reversed, leading to more smouldering and higher aerosol production. Nonetheless, we find overall reasonable agreement with AOD throughout the fire season (r<sup>2</sup> = 0.54 and RMSE = 0.05, see Figure 7a), showing that WRF captures pyrogenic aerosol trends reasonably well. While not perfect, we have found WRF to replicate the broad spatiotemporal patterns in the observations.

# 3.2 RF

For each scenario we quantified cumulative  $RF_{SLCF}$ ,  $RF_{\alpha}$ ,  $RF_{CO_2}$  and  $RF_{GHG}$  annually over a 20 year period post-fire season. Results are shown in Figure 8.

Both EDS and EDS reduced scenarios (Figure 8a & b) have a negative overall RF, both in the short-term and the long-term. On shorter time scales RF in these two scenarios could be as low as -0.006 to -0.007 W $m^{-2}$ , and in the long-term between -0.001 and -0.002 W $m^{-2}$ . In magnitude terms, this is equivalent to roughly 2-10% of RF due to global aviation emissions in

**Figure 7.** Spatially averaged AOD timeseries over the study area for (a) MODIS and (b) AERONET are shown in comparison to spatially and temporally co-located WRF AOD. The specific times and dates which are averaged over are dependent on data availability, different between MODIS and AERONET, such that the mean WRF AOD in panel (a) is averaged over a different temporal and spatial subset than panel (b) and these may therefore diverge. The blue highlighted area indicates the model spin-up. The insert in the top right of (b) shows the locations of each AERONET station. Their colour indicates the proportion of the study period for which data was available from each station, as shown in the colourbar. Times and locations where observations were unavailable in the respective observational datasets were masked in the WRF-Chem output for this figure.

2018 (Lee et al., 2021).




SLCF and total GHG RF contributions in these scenarios are negative, and the reduction in  $CO_2$  emissions is the largest single factor generating negative RF, despite the shortened lifetime.  $EF_{CO_2}$  is lower in the EDS, so proportionally less  $CO_2$  will be emitted in EDS burning scenarios. Similarly,  $RF_{SLCF}$  being negative aligns with the intra-seasonal OC emissions pattern: higher OC EFs in the EDS result in proportionally more aerosols being emitted, resulting an overall cooling effect. This also explains why the magnitude of  $RF_{SLCF}$  is lower in the EDS reduced scenario; reducing the total BA will reduce OC emissions, and with it the magnitude of  $RF_{SLCF}$ , though it will remain negative.

 $RF_{\alpha}$  in these two scenarios is substantial and positive (+0.002 to +0.003 W $m^{-2}$  in the short-term). It is the only positive component in these two EDS scenarios. Shifting burning earlier in turn causes ground to darken earlier in the season, providing more time for this lower surface albedo to contribute to the cumulative RF effect. This effect diminishes relatively quickly post fire season, however, as the vegetation will re-grow and cloud cover will persist during the wet season, effectively negating surface cover changes.

 $RF_{GHG}$  makes a relatively small but persistent negative contribution to total RF in these scenarios, around 10% of the total at the start but climbing to almost 40% after 20 years due to the longer lifetime of species involved in this component. Lower total BA reduces emissions of GHG in the EDS reduced scenario. In the EDS scenario some strong localised  $CH_4$  emissions

**Figure 8.** Cumulative RF in each scenario, split into four components: Albedo (purple), SLCF (blue), CO<sub>2</sub> (red) and other GHGs (yellow). The thick black dotted line is the sum of all the forcing components, and the thinner black dotted lines represent the upper and lower uncertainty limits. Panel (a) shows results for the EDS scenario, (b) EDS reduced, (c) LDS, (d) LDS increased and (e) EDS suppressed, over a 20-year time period. Year 0 is the year containing the fire season. Note the difference in scales between panels (a) - (d) and (e).

dynamics result in the negative  $RF_{GHG}$ , despite slightly higher  $EF_{CH_4}$  in the EDS (discussed further in section 4.1).

In the LDS and LDS increased scenario (Figure 8c & d), RF<sub>SLCF</sub> is positive and ranges from +0.002 to almost +0.004 W $m^{-2}$ . On the longer term RF from the LDS scenario is negligible, and a little under +0.001 W $m^{-2}$  in the LDS increased scenario. RF<sub> $\alpha$ </sub> is negative but smaller in magnitude than in the EDS or EDS reduced scenarios, around -0.001 W $m^{-2}$ . Later burning results in a darker land surface for a shorter period of time than EDS scenarios; the onset of the wet season means that this effect has a fairly sharp cut-off point and is thus smaller in magnitude than the EDS scenarios mentioned above.

 $RF_{CO_2}$  and  $RF_{GHG}$  are relatively small contributors in the LDS scenario - slightly negative, but other components dominate. This is likely due to a change in EFs towards those more similar to the EDS values in the very late dry season as the wet season draws closer (and thus humidity levels rise and/or fuel composition changes), as observed by Vernooij et al. (2023). In contrast  $RF_{CO_2}$  and  $RF_{GHG}$  are positive in the LDS increased scenario, mostly due to the overall increase in BA.

 $RF_{SLCF}$  is the largest single factor in both LDS scenarios, starting at just under +0.004 W $m^{-2}$  one year post-fire season. This is the opposite effect to that described in for OC emissions in the previous two scenarios; later fires mean lower OC emissions, and thus a reduced aerosol load with associated warming.

The EDS suppressed scenario has the highest uncertainty range of any scenario thanks mostly to a substantial reduction in BA and associated respiration, but also a comparatively large contribution from  $RF_{\alpha}$ . It is also the only scenario where the long-term RF could be both positive or negative, as 0 falls within the uncertainty range. Short-term RF could be as high as  $\pm 0.015 \ Wm^{-2}$  thanks to significantly lower OC emissions - both a reduction in BA and a shift to LDS burning contributing to this. However, this effect could be almost entirely compensated for by a combination of high  $RF_{\alpha}$  and possibly elevated  $CO_2$  respiration in the wet season, as shown by the lower uncertainty limit which is negative from one year post-fire season onward. Additionally, EDS suppressed is conspicuous with the highest magnitude  $RF_{SLCF}$  in any scenario. This is likely due to the double-effect of reducing the relative amount of EDS burning (which will reduce the aerosol emissions, as seen in the LDS scenario) as well as reducing total BA (further reducing overall emissions including aerosol), making this the scenario with the lowest overall aerosol emissions and thus this RF component is strongly positive.

#### 4 Discussion





Our results show that a domain-wide shift towards the EDS is generally climate-cooling, even in the long-term, while a shift towards the LDS is warming on the short-term but may not have a substantial effect after 10+ years. Reducing or increasing BA has the effect of cooling or warming respectively, in both the long and short-term. Shifting burning later but reducing BA have opposing effects on climate, as in the EDS suppressed scenario, such that the sign of RF changes from positive (warming) to negative (cooling), though the uncertainty in this case is substantial.

In the next subsection we discuss what contribution each RF component makes in each scenario. Following this, we discuss

some more localised dynamics, as well as some possible implications of this work on fire management choices in southern Africa.

#### 540 **4.1** RF

In some scenarios one component is clearly dominant (e.g  $RF_{CO_2}$  in EDS reduced or  $RF_{SLCF}$  in EDS suppressed, Figure 8b & e) whereas in others the picture is more balanced.  $RF_{GHG}$  magnitudes do not change substantially over the 20-year time period due to long lifetimes of the species involved (Prather et al., 2012, 2015), and RF components which only act until the end of the fire season or shortly after ( $RF_{\alpha}$ , and  $RF_{SLCF}$ ) diminish far quicker.  $RF_{CO_2}$  decreases at a rate somewhere between these two components, as it lasts longer than a few weeks after the fire season but still has a much shorter lifetime compared to the GHG component. These lifetime dynamics mean that is possible for short-term RF to be positive while long-term RF may be negative as in the EDS suppressed scenario (Figure 8e), though this is the only scenario where this sign flip is observed.

#### **SLCF**





Within the (relatively short) time frame of WRF-Chem simulations, aerosol cooling effects appear to be the dominant contributor to  $RF_{SLCF}$ , in line with results from Moubarak et al. (2023). It seems likely that from the cooling effects direct scattering is the more dominant, as cloud formation in the fire season is very limited over land. With more fire activity RF drops, and with less fire activity RF rises again. With the current assumption that the EFs of absorptive BC and BrC are constant across the fire season, the key determinant is then emissions of scattering OC aerosols. During the earlier part of the dry season OC emissions are proportionally higher. If burning is shifted towards the EDS then more incoming light is scattered, leading to more negative  $RF_{SLCF}$ . The assumption that  $EF_{BC}$  is constant, however, is less certain. It does not appear to change with combustion efficiency as  $EF_{OC}$  does (Vernooij et al., 2022), and we currently have no clear evidence to link changes in  $EF_{BC}$  to seasonality. This is not to say it does not vary however, as  $EF_{BC}$  can change from fire to fire (Andreae et al., 1998), but in the absence of data we must rely on a constant average value for  $EF_{BC}$ .

While this represents an important caveat in our work, the ratio of BC emissions to scattering aerosol emissions ranges typically from 1:10 to 1:5, except under extremely efficient combustion conditions (Vernooij et al., 2022) and a typical fire plume contains more OC than BC (Andreae, 2019; Bond et al., 2013). It has not been possible in this study to quantify the exact contributions of changes in fire patterns to absorptive aerosol emissions, but it seems unlikely to us that these changes would be large enough to drastically alter the conclusions presented in this paper.

These conclusions should be viewed in the context of Figure 7, which shows that WRF-Chem overestimates AOD in the early part of the fire season (around June), does reasonably well in the mid fire season, and then underestimates AOD somewhat in September. This suggests that WRF is likely to have overestimated EDS AOD and thus the cooling effect that RF<sub>SLCF</sub> has in EDS scenarios. The same may be said, albeit to a lesser extent, of the warming effect RF<sub>SLCF</sub> exhibits in LDS scenarios. This implies that, in all cases, we would expect the blue SLCF patch in Figure 8 to be smaller in magnitude. In the EDS & LDS cases (panels a-d) this is unlikely to change the long-term outcome given the dominance of GHGs or lack of appreciable

long-term impact (LDS, panel c). It is possible that for the EDS suppressed scenario, where BA is reduced exclusively in the EDS, the long-term RF may be more likely to be negative as a result of this reduction in  $RF_{SLCF}$  magnitude.

# Albedo

Direct albedo effects are relatively straightforward to model - earlier burning provides a longer period with a darker surface, and a positive  $RF_{\alpha}$ . Later burning reverses this effect, but  $RF_{\alpha}$  in these scenarios will be negated by the onset of cloud cover/green flushes at the onset of the wet season. It is therefore likely that  $RF_{\alpha}$  from scenarios with later burning will have  $RF_{\alpha}$  lower in magnitude than the EDS counterparts (as well as a sign change), as albedo changes in LDS scenarios are shorter in duration, and this is indeed what we observe.

 $RF_{\alpha}$  and  $RF_{SLCF}$  are the only two components with consistently opposite signs, though relative magnitudes do vary. These components are intrinsically linked through aerosol emissions; more BA results in more land surface with lower albedo and thus positive RF, but also in higher aerosol emissions with associated negative RF. There may be more complex interactions between albedo and scattering or possibly cloud formation that we miss in our simplified albedo parameterisation. It matters, for example, above which surface the scattering aerosols or clouds are present. If aerosols are transported above unburned surfaces then  $RF_{\alpha}$  could potentially be enhanced, and if they remain above the source region (i.e. BA) then  $RF_{\alpha}$  is similarly diminished. The type of pre-fire vegetation cover burned also affects  $RF_{\alpha}$  in a similar way. Our lack of explicit spatial analysis on  $RF_{\alpha}$  is reflected in the relatively high uncertainty on this parameter. Further examination of these spatial interactions between aerosols, vegetation cover and BA could help constrain this RF component better.

# $CO_2$






In the context of RF from savanna burning emissions,  $CO_2$  inclusion has a few complicating factors and has been excluded from much of the previous literature. This primarily on the basis that photosynthesis quickly removes such pyrogenic  $CO_2$  from the atmosphere (Lipsett-Moore et al., 2018; Landry and Matthews, 2016; Russell-Smith et al., 2021), but also from respiration dynamics (Section 2.6). Our results show that despite assumed rapid cycling from emission to sequestration via photosynthesis on the timescale of about one full year,  $CO_2$  from BB is still an important component of total RF in many scenarios, and in some cases the single largest component (EDS reduced). In this scenario  $RF_{CO_2}$  is largely driven by a reduction in total BA, but a lower EF in the EDS due to less efficient combustion also plays a part. The cause of the relative magnitude of  $RF_{CO_2}$  may be a simple matter of quantity;  $CH_4$  is 80 times stronger as a GHG than  $CO_2$  on a 20-year time horizon (IPCC, 2023), but  $EF_{CO_2}$  in the tropical savanna is usually 500-1000 times higher than  $EF_{CH_4}$  (Akagi et al., 2011; Andreae, 2019; Vernooij et al., 2023). As an aside, the  $EF_{CO_2}$  to  $EF_{BC}$  or  $EF_{OC}$  ratio is on a similar order of magnitude, though comparing the climate effects of a similar quantity of  $CO_2$  and aerosol species is far less straightforward than for  $CH_4$ .

Fire-induced CO<sub>2</sub> respiration pulses augment pyrogenic CO<sub>2</sub> emissions in burned landscapes (Sánchez-García et al., 2021). A reduction in BA and associated reduction in pyrogenic CO<sub>2</sub> also leaves more vegetation at the end of the dry season to be respired, which could compensate to some extent for this CO<sub>2</sub> reduction. These two effects are opposing in our context: more

BA in a scenario means more  $CO_2$  from post-fire pulses, but less organic material in the wet season for further respiration and vice versa. Additionally, wet season respiration rates are strongly influenced by the type of vegetation cover (Richards et al., 2012) so effects could vary substantially depending on where BA is 'lost' or 'gained' in each scenario. We have attempted to account for both of these effects in a simplified way that assumes maximum effect, manifesting in a substantial error margin for  $RF_{CO_2}$  in scenarios where there is a change in BA.

We must also consider the fact that respiration in tropical savannas can respond strongly to precipitation (Fan et al., 2015). We can imagine a situation, for example, where a prescribed burning policy is applied in a given fire year resulting in EDS shift, and the following wet season has particularly high rainfall. There is potential for  $RF_{CO_2}$  to become positive due to EDS burning if a precipitation threshold is reached, in combination with extensive litterfall in unburned areas. We already assume this litterfall to be more extensive than can be reasonably considered realistic, so we expect this rainfall threshold to be substantial for  $RF_{CO_2}$  to change sign. However, we cannot be certain, and there is likely to be spatial and temporal variability in post-fire season respiration in any burn scenario. Given that  $RF_{CO_2}$  is such a substantial contributor to total RF in some scenarios, this uncertainty warrants further study to better understand the relationship of  $CO_2$  with RF from changes to the tropical savanna fire regime.

If we exclude CO<sub>2</sub> from our analysis our long-term conclusions do not change, albeit with an RF reduced in magnitude in all cases (see Figure S7 in the supplementary material). Short-term RF does change though, most notably in the EDS reduced scenario; where it was strongly negative before, with the exclusion of CO<sub>2</sub> it becomes weakly positive (albeit with high uncertainty, and remaining negative in the long-term). In the EDS scenario the effect is similar, and we must wait at least 8 years after the fire season until we can be confident that RF in this scenario is negative, if small. For the LDS scenarios there is no such change, only the aforementioned reduction in RF magnitude. How important we consider the inclusion/exclusion of CO<sub>2</sub> may therefore depend (unsurprisingly) on the time scale of interest, and also the scenario in question. On the long-term the differences are only in magnitude, whereas the short-term the overall effect can be entirely opposite in some cases.

#### Other GHGs



Interestingly, in our simulations the changes in forcing due to  $CH_4$  and  $N_2O$  consistently share the same sign as that from  $CO_2$ . This was somewhat unexpected, given that  $EF_{CH_4}$  and  $EF_{N_2O}$  are relatively higher in the EDS and lower in the LDS, opposite to  $EF_{CO_2}$  (Vernooij et al., 2023). We had therefore anticipated that especially  $CH_4$  would have had a net positive (warming) effect in the EDS-shifted scenarios and net negative (cooling) effect in the LDS-shifted scenarios. The explanation for this not being the case lies in where and how we applied our BA shifts in each case. In some grid cells with high BA in the northwestern part of the study area (e.g. border regions between Angola, D. R. Congo and northwestern Zambia), the transition date from early to late dry season occurs early in the season. Shifting BA in these regions therefore changes little, though  $CH_4$  emissions here do increase slightly with an earlier fire regime (Figure 9a). In the eastern part of the domain (northeastern Zambia and Mozambique) the transition dates are far later, and in fact  $EF_{CH_4}$  around this time is more likely to be higher than during the

**Figure 9.** Difference in total fire season CH<sub>4</sub> emissions between the baseline and (a) EDS, (b) EDS reduced, (c) LDS, (d) LDS increased and (e) EDS suppressed scenarios, in tonnes per 30km grid cell. Positive values indicate areas that produce comparatively more CH<sub>4</sub> over the fire season, and negative values indicate areas where CH<sub>4</sub> emissions are reduced.

peak fire season (Vernooij et al., 2023). Shifting burning earlier in these regions means that less burning occurs under these conditions which is conducive to higher  $CH_4$  emissions. This effect is fairly local, but still strong enough to dominate  $RF_{CH_4}$  when averaged over the entire domain.

# 4.2 Implications for fire regime changes

This paper focuses on the impacts of fire regime change on RF, but we must acknowledge that there are many other factors to consider when deliberately using fire in a landscape. Other established uses of fire can both be complementary or detrimental to reducing RF (Butz, 2009; Huffman, 2013; Garde et al., 2009; Russell-Smith et al., 2021). We do not discuss these many other uses of fire here, but our conclusions on the subject of fire regime change should be taken together with other relevant aspects of burning in a given locality, especially if being used to inform decisions on when and where burning is desirable.

#### **Transition dates**






We have chosen to use spatially variable transition dates based on the relative increase of night-time burning (Eames et al., 2023). As explained in section 2.2, these transition dates determine how much BA is available for shifting, and choosing them differently could impact the BA distribution in our scenarios. We believe this choice is justified for our purpose as prescribed fires are ideally not permitted to continue into the night, and the resulting BA distribution from each scenario represents perhaps an extreme, albeit not unrealistic (see supplementary material). We have made efforts to ensure that BA shifts only occur in areas with sufficient fire activity, and align with the part of the season that is feasible for such burning. We recognise that alternative methods of defining a transition date may be possible, though we do not expect this to alter our conclusions significantly. This is because a more extreme shift than currently modeled in EDS and LDS scenarios would be unfeasible. A smaller BA shift may reduce the magnitude of our various RFs, but is unlikely to change their sign or the relative sizes. Furthermore, the use of these spatially dynamic transition dates is far more representative of seasonal fire dynamics in southern African savannas compared to using a fixed transition date. Other methods such as using the driest month of the year (Lipsett-Moore et al., 2018) also may miss some of the local fuel dynamics which determine suitability for prescribed fire.

#### 660 Location matters

One key implication (illustrated in Figure 9) is that the location of burning may be just as critical as the timing when considering changes to the fire regime. Negative  $RF_{GHG}$  in the EDS scenario, for example, is largely achieved by  $CH_4$  emissions reduction in two areas in the eastern part of the domain (northeastern Zambia and the northern half of Mozambique). Elsewhere, an EDS burning shift generally leads to less pronounced changes, or even increases in  $CH_4$  emissions. If a reduction in BA can be achieved along with the timing shift (i.e. the EDS reduced scenario) then  $CH_4$  emissions decrease in a larger portion of the domain, further contributing to the RF reduction. Still, even with reduced BA some grid cells still show higher  $CH_4$  emissions in the EDS reduced scenario (northwestern corner of Figure 9b). This is important for local land management organisations to take into consideration when opting for changes in fire regimes. This is also not limited to  $CH_4$ , as  $N_2O$  and NMVOC emissions follow a similar dynamic, and for similar reasons, as discussed in the previous section.

Aerosols are an important contributor to  $RF_{SLCF}$ , such that the fuel type and condition must also be taken into consideration when assessing if a given location is suitable for fire regime changes. Aerosols are more readily produced from smouldering combustion and/or less efficient burning (Andreae et al., 1998), which tends to be the case in areas with a higher proportion

of woody fuel available. However, smouldering also produces more  $CH_4$  (Vernooij et al., 2022). Aerosol emission may be counterbalanced by the emission of warming GHGs, especially in the longer term (as seen in Figure 8e). If the aim is to change the fire regime in a given location to reduce overall forcing, attention must be paid to the local  $EF_{OC}$  in comparison to GHGs such as  $CH_4$ .

#### Other impacts related to fire regime change







Aerosols are linked to detrimental health issues in humans (Reid et al., 2016; Chen et al., 2021) and in animals (Sanderfoot et al., 2021). In an ecosystem which has such a strong relationship with fire (Beerling and Osborne, 2006), changes to fire patterns may also have consequences for flora in the region. Positive impacts may include increased biodiversity (e.g. Evans and Russell-Smith, 2020), while impacts such as bush encroachment may be less desirable (e.g. Case and Staver, 2017). Whether a change in fire regime is suitable for a given location may depend on these and other local factors, all of which should be given due consideration.

Social aspects are affected too, as burning is deeply rooted in culture and tradition in the savanna (Sluyter and Duvall, 2016). Here, there is a potential income stream for indigenous populations via the use of prescribed fire for climate cooling and/or carbon offsetting. This has already been achieved to a certain extent in Australian tropical savannas (Russell-Smith et al., 2013; Russell-Smith et al., 2021). If done thoughtfully, these projects can represent a win-win-win for the climate in terms of (slightly) negative RF, for the ecological health of landscapes in relation to desirable burning, and for local people on what is often (and given the outcomes of this study perhaps unjustly) thought of as 'unproductive' land. However, clearly a RF reduction equivalent of about 10% of global aviation emissions is no magic bullet in climate change terms, and we especially want to avoid encouraging offset activity in the global south to justify a 'business as usual' future emissions scenario.

#### **Outlook for future burning projects**

As shown in Figure 9, some few key locations may drive a substantial proportion of RF, such that widespread alterations to a fire regime are not necessary to achieve maximum effect. It is difficult to pin down exactly where these locations might be as (unlike GHGs) aerosol forcing can vary substantially depending on e.g. cloud cover, land cover type and/or atmospheric transport patterns (Bellouin et al., 2020). We therefore urge caution in interpreting this paper as grounds to advocate for general fire regime change in southern Africa. The core message we wish to convey is that once all major RF factors are taken into account, **overall** the GHG emissions outweigh the climate effects of surface albedo changes and SLCF contributions.

This message should be viewed within the context of other fire management challenges and outcomes. Our scenarios represent an idealised case, where all burning in the southern African savannas is conducted with a view to affecting RF. We find that there may be some locations where this is justified in this narrow perspective, but do not attempt to reconcile this with other management goals or take into account practical limitations. There is more work necessary to better understand how burning for climate benefit may fit into the wider fire management landscape in southern Africa (Knowles et al., 2025). Future burning projects should be delivered hand-in-hand with ecological and socially positive outcomes to be fully justified.

#### 5 Conclusions




We expanded on previous studies focused on net GHG emissions from landscape fires to include effects of aerosols and other SLCFs, surface albedo changes, and  $CO_2$ . The inclusion of  $CO_2$  in particular is important, as we showed that despite assuming a short lifetime and enhanced respiration effects  $RF_{CO_2}$  can be substantial contributor to net cooling or warming. Notwith-standing the complexities and associated uncertainty  $CO_2$  brings, it is clearly a major factor in the climate effects of biomass burning, and we strongly advocate for its inclusion in any future studies. We also believe that the availability of better data on the seasonality of absorptive aerosol (BC and BrC) EFs would improve this and any future work on this subject, and this currently represents a key caveat in our conclusions.

Though our study includes several additional components, we find ourselves agreeing with previous studies limited to net GHG emissions (e.g. Lipsett-Moore et al., 2018; Russell-Smith et al., 2021) that a shift to earlier burning patterns generally results in a climate cooling effect.

We have demonstrated that changing the timing of fires in the southern African savanna can impact global climate via several key RF components. Short-lived effects (surface albedo changes and SLCF) dominate in the short term, but longer-term the longer-lived species of GHG are more important. The magnitude of RF is always greater in the short term than in the long term, e.g.  $-0.004 \text{ W}m^{-2}$  goes to  $-0.002 \text{ W}m^{-2}$  in the EDS reduced scenario or  $+0.003 \text{ W}m^{-2}$  goes to  $+0.001 \text{ W}m^{-2}$  in the LDS increased scenario. However, we can be confident that RF from most scenarios is consistently either positive (LDS shift) or negative (EDS shift) on any time scale. Exceptions to this are the EDS suppressed scenario, or if the effects of  $CO_2$  are excluded from the EDS reduced scenario. In magnitude terms, both EDS and LDS shifts are comparable to around 2-10% of RF from commercial aviation in 2018.

This study has taken a broad view of a single fire season across the southern African savanna region. However, location of BB matters as well as timing, and careful choices in where to burn and when to burn could enhance the cooling or warming effects of our scenarios. We showed that shifting burning patterns earlier in areas with a heavily late-skewed fire season can substantially reduce e.g. CH<sub>4</sub> emissions from these areas. Being more selective in where the timing of fire is altered in this way could be a more efficient method to achieve a climate cooling effect. With our holistic view of the southern African savanna fire system and despite an occasionally large uncertainty in the magnitude of the effect, we conclude that on a continental scale, shifting to more early season burning is climate cooling, and more later burning is climate warming. At the last, we repeat our note of caution on using this study to justify broad changes to southern African fire regimes; aside from wariness of specific local feedback, such as a damping of RF due to surface albedo changes as well as more pyrogenic aerosol scattering above a burn scar. We must also consider whether such fire regime change is to the benefit of the local ecosystem and population, else risk creating more problems in our pursuit of the solution of another.

Data availability. GFED5 burned area data is available at https://doi.org/10.5281/zenodo.7668423

Tree biomass data is available on request from Alexandre Bouvet (alexandre.bouvet@cesbio.cnes.fr)

Other fuel load data is available on request.

Data used to build the EF model output used is available from https://doi.org/10.5281/zenodo.7689032. The EF data itself is available on request to Roland Vernooij (roland.vernooij@wur.nl)

Meteorological data used to drive the WRF-Chem model runs is available from the NCAR repository: https://rda.ucar.edu/datasets/d083002/dataaccess/#

Tailored scenario burned area and emissions data used to drive WRF-Chem simulations are available at https://zenodo.org/records/15578063

CAMS anthropogenic and biogenic emissions data is available from the ECMWF atmosphere data store: https://doi.org/10.24381/d58bbf47

Biomass burning emissions from GFED4.1s used in biomes outside the southern African savanna can be found at https://www.geo.vu.nl/~gwerf/GFED/GFED4/

CAM-chem model output for WRF-chem initial and boundary conditions can be found at https://www.acom.ucar.edu/cam-chem/cam-chem.

shtml

TROPOMI Sentinel-5P data are available on the Copernicus data store: https://dataspace.copernicus.eu/explore-data/data-collections/sentinel-data/sentinel-5p

AERONET version 3 station data can be downloaded from https://aeronet.gsfc.nasa.gov/new\_web/webtool\_aod\_v3.html MODIS MAIAC AOD data can be downloaded from https://lpdaac.usgs.gov/products/mcd19a2v061/

Author contributions. TE: Conceptualization, Methodology, Software, Investigation, Formal analysis, Writing - Original Draft, Visualisation

NS Conceptualization, Investigation, Methodology, Software, Writing - Review & Editing

EI: Formal analysis, Methodology, Software, Writing - Review & Editing

IvdV Conceptualization, Methodology, Software, Writing - Review & Editing

MJvG Investigation, Methodology, Software, Writing - Review & Editing

RV Software, Methodology, Formal analysis, Writing - Review & Editing

GRvdW: Conceptualization, Formal analysis, Writing - Review & Editing, Funding acquisition, Supervision

Competing interests. The authors declare no conflict of interest.

Acknowledgements. This work was carried out with the support of the 2017 Ammodo Science Award for Natural Sciences. The authors are grateful for the use of the BAZIS HPC cluster made available at the VU Amsterdam. We also thank SURF (www.surf.nl) for the support in using the National Supercomputer Snellius. We acknowledge use of the WRF-Chem preprocessor tool mozbc provided by the Atmospheric Chemistry Observations and Modeling Lab (ACOM) of NCAR.

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
