# Peer review of "Radiative forcing due to shifting southern African fire regimes"

_EGUsphere, 2025_

## Referee Comment (RC2)

Review of "Radiative forcing due to shifting southern African fire regimes" by Tom Eames et al.

This study aims to provide an assessment of how changing fire regimes in southern African savannas affects global climate forcing. Moving beyond previous research that only examined $CH_4$ and $N_2O$ emissions, the researchers evaluate the complete climate impact of prescribed early-season burning versus late-season fires by incorporating $CO_2$ emissions, aerosols, short-lived climate forcers, and surface albedo changes. WRF-Chem model is used as a primary tool to calculate the radiative forcing (RF) impacts due to aerosols and other short-lived climate forcers, while $RF_{GHGs}$ are calculated separately. Their findings indicate that shifting fires earlier in the dry season generally produces a cooling effect (negative radiative forcing) of approximately -0.001 to -0.006 $Wm^{-2}$, with $CO_2$ emission reductions and albedo effects being major contributors. Conversely, late season burning tends to create a warming effect of smaller magnitude. The research aims to inform emissions mitigation programs currently operating in Australia and expanding into Africa, while emphasizing that the climate benefits of fire regime modifications must be evaluated carefully at local scales due to significant regional variations in effectiveness.

The study claims to be a novel effort in quantifying the impact of fire regime changes on Earth's radiative forcing, which is valuable to the scientific community and within the scope of ACP. However, some major clarifications are required regarding set-up of modeling experiments and calculations of Radiative Forcing (RF).

Major Comments:

1. Further clarifications are required for the need to present the RF estimates as a global average rather than a regional average as calculated by WRF-Chem over the southern African domain. The projection or extrapolation of regional RFs to global scale could have been a section in addition to presenting regional mean estimates and distribution over the domain (which one would hope to have higher and more significant magnitudes). If indeed global estimates were the intended outcome, why use WRF-Chem as the tool for such a study and not use a global model rather to avoid the additional uncertainties in projection/extrapolation?

2. In section 2.1, authors say, "In this paper we only address changes to global RF because of shifts in southern African fire patterns, and all numbers presented should be interpreted in this context. We do not concern ourselves with the absolute forcing as a result of tropical fires as a whole, but rather the difference relative to a pre-defined baseline scenario." Is the baseline scenario one of the WRF-Chem experiments? If so, please include or call it out in Table 1. Further details and

equations are required throughout section 2.3 and 2.4 to explain how RF of each component (aerosols, albedo change etc.) are extrapolated to a global scale and made commensurate with $RF_{GHG}$ when presented together in Figure 8?

3. The model experimental set-up is confusing in terms of the choice of initial and boundary conditions. In Section 2.3.2, it says that Gas, NMVOC and aerosol initial and boundary conditions were adapted from CAM-Chem (Buchholz et al., 2019). When repeating model experiments for each year after 2019 for the seven-month period, were the "Gas and NMVOCs" (including CH4, N2O, and NMVOCs) initial and boundary conditions reset to 2019 values at the start of April every year? Please explain how the continuity of gas concentrations is maintained for the 20 years following the baseline year of 2019 to be able to calculate reasonable $RF_{GHG}$ changes?

4. The AOD comparisons of WRF-Chem to MODIS and AERONET both show a substantial overestimation of model AOD during June month, which is also the peak of your EDS (Fig. 2a). Please add a discussion on how this uncertainty should impact the magnitudes of your results for RF changes due to aerosols.

5. Section 4.2 is long-winded and hard to follow. Consider making it concise labeling the sub-sections as well. More importantly, "location matters" subsection within 4.2 highlights the criticality of location of burning versus timing of burning and that the climate benefits of fire regime modifications must be evaluated carefully at local scales due to significant regional variations in effectiveness. Therefore, it would be valuable to complement this discussion with spatial maps of the region showing the magnitudes of RF changes for each component (GHG, aerosol and surface albedo) rather than just detailing in words that is hard to visualize and follow.

Specific Comments:

▪ Line 29-31: "The optical properties of pyrogenic aerosols, secondary aerosol effects (such as aerosol-cloud interactions…". There are specific terms describing these effects, called the "direct, indirect and semi-direct effects of aerosols". It could also be rephrased as Aerosol-radiation interactions (ARI) and Aerosol-cloud interactions (ACI) to be consistent with IPCC. Consider revising this sentence to include either of these set of terms to define the effects of aerosols. Moreover, please include suitable references that studied ARI and ACI, especially over the region of interest. The phrase "…well documented climate warming effect from (pyrogenic) GHGs" prior to only GHGs makes it sound like aerosol effects aren't documented, so consider dropping this phrase completely.

- Line 36-37: "...aerosol-climate interactions are less certain than those attributable to GHGs". Consider adding a sentence or two to explain why so?
- Line 44-46: Emission proportions are also heavily dependent on "type of burning" (i.e., flaming versus smoldering). Add that to this sentence with suitable refernces.
- Lie 264-65: Why are the other BB emissions from GFED4 while BA is from GFED5?

**Editorial Comments:**

Line 51: remove "e.g."

Fig. 2a: There is no labeling of the y-axis. Please add that.

Fig. 2b: Suggest changing the three different colored blue lines to "distinct" colors (e.g., RGB) for better legibility.

---

## Author Comment (AC1)

The authors wish to thank Oliver Perkins and two anonymous reviewers for their comments on this manuscript. We hope to have sufficiently addressed all concerns raised in those comments.

Our response to the comments below are in blue, and any passages from the updated manuscript are shown in red.

**O. Perkins**

Thank you to the authors for this interesting paper. I am basically supportive of what you are doing here. I appreciate that your main focus is atmospheric chemistry, and there you produce some very interesting results. However, I have some clarifying questions about the human dimensions of your scenarios. I think their presentation should be tweaked to guard against potential misuse, e.g., in support of unsuitable savanna abatement schemes.

(1) Human impact on the baseline scenario

I think you need to do more to recognise that in your study region, human controlled burning already plays a significant role in the fire regime. It is good you recognise that EDS burning is present in many parts of the study region. However, it is not always as simple as human EDS burning vs LDS wildfires.

For example, in North-eastern Zambia, which has a significant impact on your results, there is widespread fire use for shifting cultivation1,2,3. This often occurs very late in the dry season, just before the first rains. This then, is a very different form of fire use from early dry-season burning: it either occurs before the first rains, or it doesn't make sense from an agricultural point of view. Hence, it is not that surprising that pushing this into the early dry season has a substantial impact on RF, but I'm not sure of the real world applicability of this finding.

(2) Human implications of shifting fire use

I think this difficulty may arise from an overreliance on the Australian case study as a conceptual model. There, Aboriginal fire was systematically removed from the landscape, before its targeted reintroduction. As such, a targeted early burn was possible. This is not the case in your study region, where fire use is a fundamental component of peoples' livelihoods, and often their social/religious identities, and occurs at many points throughout the dry season for different reasons. Hence, intervening in the way implied by your scenarios is likely to be much more complicated than in the Australian case [see 4 for a real world example of what happens when this goes wrong].

It is good that you note the possibility of local/regional scale difficulties in implementing your scenarios. However, the section beginning at line 674 should be rewritten to reflect the different context in which your scenarios are being run to Northern Australia, and do more to highlight they are theoretical maximums that would have very substantial feasibility challenges for real world application. Some more engagement with the specifics of the study context would be useful. See 5 for a global synthesis / index of literature on human fire use, from where I located refs 1,2,3.

Thank you again for the interesting work.

Oliver Perkins

- 1 https://repository.kulib.kyoto-u.ac.jp/items/02ea39b5-5ea9-4571-a37c-12afc3ca5f0b
- 2 https://www.tandfonline.com/doi/full/10.1080/00380768.2014.883487
- 3 https://www.jstor.org/stable/30135835#metadata\_info\_tab\_contents
- 4 https://agupubs.onlinelibrary.wiley.com/doi/10.1029/2023EF003552
- 5 https://figshare.com/collections/DAFI\_a\_global\_database\_of\_Anthropogenic\_Fire/5290792/4

We thank O. Perkins for their insightful comments on our work. As they rightfully point out, the contexts of Australia and southern Africa are fundamentally different in some key ways, affecting the viability of such burning projects as discussed in the manuscript. The recent paper by Knowles et al. (2025, <a href="https://doi.org/10.1038/s41893-024-01490-9">https://doi.org/10.1038/s41893-024-01490-9</a>) is a neat summary of these challenges. To better acknowledge this, and to emphasise that our paper represents but one part of the fire management puzzle in southern Africa, we have added the following text:

**Introduction (Lines 80-85):**

While the success of the Australian project is the motivating factor for this southern African study, it is also important to acknowledge that there are different considerations for the African continent. These considerations (including biodiversity goals and impacts on ecosystems and livelihoods) are detailed in Knowles et al. (2025), where the authors conclude that more evidence is required to show that a shift in fire timing will reduce greenhouse gas emissions. This work is an attempt rectify this, focusing on the climate impacts of such fire regime change to add to the discussion on whether such burning is desirable in the larger social, ecological and financial contexts of southern Africa.

Discussion (section 4.2, "Outlook for future burning projects", lines 695 - 706):

As shown in Figure 9, some few key locations may drive a substantial proportion of RF, such that widespread alterations to a fire regime are not necessary to achieve maximum

effect. It is difficult to pin down exactly where these locations might be as (unlike GHGs) aerosol forcing can vary substantially depending on e.g. cloud cover, land cover type and/or atmospheric transport patterns (Bellouin et al., 2020). We therefore urge caution in interpreting this paper as grounds to advocate for general fire regime change in southern Africa. The core message we wish to convey is that once all major RF factors are taken into account, overall the GHG emissions outweigh the climate effects of surface albedo changes and SLCF contributions.

This message should be viewed within the context of other fire management challenges and outcomes. Our scenarios represent an idealised case, where all burning in the southern African savannas is conducted with a view to affecting RF. We find that there may be some locations where this is justified in this narrow perspective, but do not attempt to reconcile this with other management goals or take into account practical limitations. There is more work necessary to better understand how burning for climate benefit may fit into the wider fire management landscape in southern Africa (Knowles et al., 2025). Future burning projects should be delivered hand-in-hand with ecological and socially positive outcomes to be fully justified.

**Reviewer 1**

Eames et al. present a comprehensive and well-structured analysis of the climate impacts associated with deliberate modifications to fire regimes in Southern Africa. The study extends beyond the roles of  $CH_4$  and  $N_2O$  to also account for aerosols, other short-lived species,  $CO_2$ , and surface albedo changes. A range of burning scenarios is investigated, and their corresponding radiative forcings are assessed using WRF-Chem model simulations. The model outputs are further evaluated against both ground-based observations and satellite datasets. Overall, the results are clearly presented, and the work fits well within the scope of ACP. Few comments are as follows:

**Major Comments:**

- 1. The manuscript is generally quite lengthy, particularly the Methodology section. The authors may consider moving various details (e.g. WRF-Chem model details including Appendix C, selected figures from the Appendix, etc......) to the supplementary material. This would improve readability, streamline the manuscript, and allow greater emphasis on the Results and Discussion sections.
  - We agree with this comment and now provide a separate supplement as suggested. Additionally we have made some structural changes to some parts of the discussion for readability, see our response to reviewer 2's comment 5 below.
- 2. The manuscript should be refined in its wording to more explicitly address the real-world implications of shifting burning to the early dry season (EDS), including potential adverse

consequences. In addition, the authors should carefully discuss the extent to which the findings from EDS burning in this study may, or may not, be applicable to other fire-prone regions globally. It should also be made clear how and to what extent these results can be referenced, and where caution is required in their broader application.

We fully agree that there is a need to prevent the potential misuse of our conclusions to implement inappropriate changes to fire regimes. We have added text to the introduction and discussion to address this, please see our response to O. Perkins above.

**Minor suggestions:**

1. Lines 10-11: It will be good to mention quantitative how much warming is produced by shifting burning to late fire season.

We have added the following sentence to the abstract:

Forcing from shifting burning later in contrast is negligible in the long term.

2. Figure 2: It would be helpful to further clarify Figure 2a, specifically what the y-axis represents and how the different peaks in EDS, LDS, and the mean should be interpreted.

Figure 2 has been edited accordingly, see our response to editorial comments at the end of this document.

3. Figure 6: Since WRF-Chem provides AOD data in regions where MODIS observations are unavailable, it would be useful to show a continuous AOD distribution in Figure 2b.

This figure is designed as a comparative exercise with MODIS AOD observations. MODIS observations are spatially limited as the reviewer notes, and are also only obtained at certain times of day. Comparing these spatially and temporally limited observations to the full WRF-Chem domain and time period may therefore misrepresent the spatial agreement of MODIS observations and WRF-Chem output. We have adjusted the figure caption to clarify this:

Figure 6. 550nm AOD for the study area from (a) MODIS MCD19A2 product and (b) WRF-Chem from mid-April to the end of October. The WRF-Chem output has been temporally and spatially co-located with the MCD19A2 product to best represent the comparison between the two. White areas indicate regions of no data in the MCD19A2 product.

4. Figure 7: For panel (a): Since MODIS data are available only once per day, is the WRF-Chem data averaged over the same daily period near the MODIS overpass time? Additionally, in areas where MODIS data are unavailable, as shown in Figure 6, are these regions excluded from the WRF-Chem averaging? For panel (b): Is the WRF-Chem averaging performed across the AERONET sites? What temporal resolution of AERONET AOD data is used here, and is the WRF-Chem averaging done at the same temporal resolution?

As noted above, WRF-Chem output is temporally and spatially co-located with both MODIS and AERONET data. AERONET data is sometimes available only sporadically,

though usually there is a WRF-Chem output datapoint close enough to any available AERONET data (WRF output frequency is every 3 hours).

Figures 6 & 7 therefore represent the level of agreement between WRF-Chem & observational datasets *only where these observations are available*. Datapoints in the WRF-Chem output that did not correspond to an observational datapoint were masked from this process. We have added the following line to the caption of Figure 7:

Times and locations where observations were unavailable in the respective observational datasets were masked in the WRF-Chem output for this figure.

**Reviewer 2**

This study aims to provide an assessment of how changing fire regimes in southern African savannas affects global climate forcing. Moving beyond previous research that only examined CH4 and N2O emissions, the researchers evaluate the complete climate impact of prescribed early-season burning versus late-season fires by incorporating CO2 emissions, aerosols, short-lived climate forcers, and surface albedo changes. WRF-Chem model is used as a primary tool to calculate the radiative forcing (RF) impacts due to aerosols and other short-lived climate forcers, while RFGHGs are calculated separately. Their findings indicate that shifting fires earlier in the dry season generally produces a cooling effect (negative radiative forcing) of approximately -0.001 to -0.006 Wm-2, with CO2 emission reductions and albedo effects being major contributors. Conversely, late season burning tends to create a warming effect of smaller magnitude. The research aims to inform emissions mitigation programs currently operating in Australia and expanding into Africa, while emphasizing that the climate benefits of fire regime modifications must be evaluated carefully at local scales due to significant regional variations in effectiveness. The study claims to be a novel e ort in quantifying the impact of fire regime changes on Earth's radiative forcing, which is valuable to the scientific community and within the scope of ACP. However, some major clarifications are required regarding set-up of modeling experiments and calculations of Radiative Forcing (RF).

**Major Comments:**

1. Further clarifications are required for the need to present the RF estimates as a global average rather than a regional average as calculated by WRF-Chem over the southern African domain. The projection or extrapolation of regional RFs to global scale could have been a section in addition to presenting regional mean estimates and distribution over the domain (which one would hope to have higher and more significant

magnitudes). If indeed global estimates were the intended outcome, why use WRF-Chem as the tool for such a study and not use a global model rather to avoid the additional uncertainties in projection/extrapolation?

We agree with the reviewer that extrapolation from regional model to a global scale introduces a degree of uncertainty that we would rather avoid. However, the aim of this study was to quantify the effects of changes on a finer spatial scale, and especially to highlight the different outcomes the same fire regime changes can have locally. This influenced our choice of model. Additionally, while nested grids are an option within WRF-Chem, the computational expense of a global run with a southern African nested grid on a sufficiently high resolution was prohibitive with our resources. The extrapolation approach outlined in the manuscript was therefore the best approach available.

We have clarified the precise methods we use in this extrapolation, see our response to the following comment.

2. In section 2.1, authors say, "In this paper we only address changes to global RF because of shifts in southern African fire patterns, and all numbers presented should be interpreted in this context. We do not concern ourselves with the absolute forcing as a result of tropical fires as a whole, but rather the difference relative to a predefined baseline scenario." Is the baseline scenario one of the WRF-Chem experiments? If so, please include or call it out in Table 1. Further details and equations are required throughout section 2.3 and 2.4 to explain how RF of each component (aerosols, albedo change etc.) are extrapolated to a global scale and made commensurate with RFGHG when presented together in Figure 8?

The baseline scenario is included in table 1 as the 'Mean' scenario. To make this clearer, we have renamed it Mean (baseline) in the table.

For albedo, Dintwe et al. (2017) use the ratio of burned area in southern hemisphere Africa to total land surface area globally to scale their regional forcing estimate to a global one. We follow this example and have updated the text to describe the process. Lines 351-354 now read:

Dintwe et al. (2017) determined that in southern hemisphere Africa, average regional RF over the fire season due to fire-induced albedo changes is around +0.33 Wm-2. They use the ratio of total burned area to global land surface area to scale this regional forcing to a global one. Doing the same using the total BA in the Mean (baseline) scenario, we arrive at a global forcing of +0.046 Wm-2.

The procedure for SLCFs is similar. Lines 300-301 have been added to clarify:

By using the ratio of the model domain area to total global surface area we may scale WRF-Chem regional forcing up to a global RF estimate, accounting for outflow across the model boundary.

3. The model experimental set-up is confusing in terms of the choice of initial and boundary conditions. In Section 2.3.2, it says that Gas, NMVOC and aerosol initial and boundary conditions were adapted from CAM-Chem (Buchholz et al., 2019). When repeating model experiments for each year after 2019 for the seven-month period, were the "Gas and NMVOCs" (including CH4, N2O, and NMVOCs) initial and boundary conditions reset to 2019 values at the start of April every year? Please explain how the continuity of gas concentrations is maintained for the 20 years following the baseline year of 2019 to be able to calculate reasonable RFGHG changes?

The long-term  $RF_{GHG}$  calculations and WRF-Chem runs are two separate things, and we have subtracted  $RF_{GHG}$  from the WRF simulations to ensure that RFWRF only accounts for SLCFs and not GHGs (i.e.  $RF_{WRF} = RF_{SLCF}$ ). We only run WRF-Chem for a single year.

 $RF_{GHG}$  therefore does not rely on the initial and boundary conditions the reviewer mentions above. This calculation instead uses SSP2-4.5 GHG projections and a simple parameterisation based on Moubarak et al. (2023) and Etminan et al. (2016), which is outlined in section 2.5.

To ensure this is clear, we have added the following to the text:

**Lines 228-233:**

The total forcing from these species is calculated on-line within each WRF-Chem scenario simulation, itself then compared to a baseline model run to calculate the total RF for that particular scenario in a single fire year. Long-lived GHGs also affect forcing within the WRF-Chem simulation, and to avoid double-counting we subtract this RF component from the total WRF-Chem RF to obtain RF solely from short-lived species (forcing from long-lived GHGs is separate to the WRF-Chem simulation and outlined in section 2.5). The contribution from SLCFs on the longer term represents the cumulative RF from the WRF-Chem simulation period, and the contribution from that specific fire season over the following period (which is zero).

**Lines 246-247:**

We ran the WRF-Chem model for a single seven-month period within each scenario using meteorology, initial and boundary conditions from the year 2019.

Details on the RFGHG calculation are then found in lines 365-370:

To include long-term GHG forcing in our analysis of possible fire regime changes, we adapted the methods outlined in Moubarak et al. (2023) for the savanna biome, using emissions from the different fire scenarios as described in section 2.3.2. For ambient

GHG concentrations, we assumed the Earth to be following a moderate climate mitigation trajectory as in the 'middle of the road' Shared Socio-economic Pathway (SSP) 2-4.5 (IPCC, 2023; Meinshausen et al., 2020) in our RF calculation. We include relevant long-lived pyrogenic species: CH4, N2O, and other NMVOCs.

4. The AOD comparisons of WRF-Chem to MODIS and AERONET both show a substantial overestimation of model AOD during June month, which is also the peak of your EDS (Fig. 2a). Please add a discussion on how this uncertainty should impact the magnitudes of your results for RF changes due to aerosols.

This is an important point and we thank the reviewer for raising it. We have added a paragraph to section 4.1 under 'SLCF' addressing this. Lines 567-573 now read:

These conclusions should be viewed in the context of Figure 7, which shows that WRF-Chem overestimates AOD in the early part of the fire season (around June), does reasonably well in the mid fire season, and then underestimates AOD somewhat in September. This suggests that WRF is likely to have overestimated EDS AOD and thus the cooling effect that RF $_{SLCF}$  has in EDS scenarios. The same may be said, albeit to a lesser extent, of the warming effect RF $_{SLCF}$  exhibits in LDS scenarios. This implies that, in all cases, we would expect the blue SLCF patch in Figure 8 to be smaller in magnitude. In the EDS & LDS cases (panels a-d) this is unlikely to change the long-term outcome given the dominance of GHGs or lack of appreciable long-term impact (LDS, panel c). It is possible that for the EDS suppressed scenario, where BA is reduced exclusively in the EDS, the long-term RF may be more likely to be negative as a result of this reduction in RF $_{SLCF}$  magnitude.

5. Section 4.2 is long-winded and hard to follow. Consider making it concise labeling the sub-sections as well. More importantly, "location matters" subsection within 4.2 highlights the criticality of location of burning versus timing of burning and that the climate benefits of fire regime modifications must be evaluated carefully at local scales due to significant regional variations in effectiveness. Therefore, it would be valuable to complement this discussion with spatial maps of the region showing the magnitudes of RF changes for each component (GHG, aerosol and surface albedo) rather than just detailing in words that is hard to visualize and follow.

We have removed some unnecessary text in section 4.2 and added in an extra subheadings (Transition dates & Outlook for future burning projects) as suggested.

Creating maps explicitly showing where RF is generated is not trivial, which we clarify in lines 695 – 698:

As shown in Figure 9, some few key locations may drive a substantial proportion of RF, such that widespread alterations to a fire regime are not necessary to achieve maximum effect. It is difficult to pin down exactly where these locations might be as (unlike GHGs)

aerosol forcing can vary substantially depending on e.g. cloud cover, land cover type and/or atmospheric transport patterns (Bellouin et al, 2020).

However, we agree that more information can be presented in Figure 9, and have extended it to include all scenarios examined. This updated figure now looks like this:

Figure 9: Difference in total fire season CH\$\_4\$ emissions between the baseline and (a) EDS, (b) EDS reduced, (c) LDS, (d) LDS increased and (e) EDS suppressed scenarios, in tonnes per 30km grid cell. Positive values indicate areas that produce comparatively more CH\$\_4\$ over the fire season, and negative values indicate areas where CH\$\_4\$ emissions are reduced.

We have created additional figures for other important species as well as the spatial distribution of the albedo forcing. These will be available in the supplementary material:

Figure S8: Spatial distribution of RF due to change in surface albedo in (a) EDS, (b) EDS reduced, (c) LDS, (d) LDS increased and (e) EDS suppressed scenarios.

Figure S9: Spatial distribution of changes in  $CO_2$  emissions in (a) EDS, (b) EDS reduced, (c) LDS, (d) LDS increased and (e) EDS suppressed scenarios.

Figure S10: Spatial distribution of changes in OC aerosol emissions in (a) EDS, (b) EDS reduced, (c) LDS, (d) LDS increased and (e) EDS suppressed scenarios.

**Specific Comments:**

Line 29-31: "The optical properties of pyrogenic aerosols, secondary aerosol effects (such as aerosol-cloud interactions....". There are specific terms describing these effects, called the "direct, indirect and semi-direct effects of aerosols". It could also be rephrased as Aerosol-radiation interactions (ARI) and Aerosol-cloud interactions (ACI) to be consistent with IPCC. Consider revising this sentence to include either of these set of terms to define the effects of aerosols. Moreover, please include suitable references that studied ARI and ACI, especially over the region of interest. The phrase "...well documented climate warming effect from (pyrogenic) GHGs" prior to only GHGs makes it sound like aerosol effects aren't documented, so consider dropping this phrase completely.

This sentence has been revised as suggested. It now reads:

Aerosol-radiation interactions (ARI; Carter et al., 2021; Canut et al., 1996), aerosol-cloud interactions (ACI; Tosca et al., 2014; Logan et al., 2024), climate warming effect from (pyrogenic) GHGs (IPCC, 2023; Etminan et al., 2016; Meinshausen et al., 2017), mean that BB emissions can affect the global climate in complex ways.

Line 36-37: "...aerosol-climate interactions are less certain than those attributable to GHGs". Consider adding a sentence or two to explain why so?

**Done. Lines 31-35 now read:**

...aerosol-climate interactions are less certain than those attributable to GHGs (Carslaw et al., 2010; Forster et al., 2021). Uncertainty in ARI is driven by several factors, including uncertainty in the underlying mechanisms, but also by a large spread in multimodel ensembles used to quantify it (Peace et al., 2020). ACI uncertainty is contained mostly in the uncertainty of the 'susceptibility' of cloud droplet number to aerosol load (Gryspeerdt et al., 2023).

Line 44-46: Emission proportions are also heavily dependent on "type of burning" (i.e., f laming versus smoldering). Add that to this sentence with suitable references.

**The sentence now reads:**

The proportion of each of these climate-influencing species within a given fire plume depends on a number of factors including the type of vegetation being burned, the condition of that vegetation (e.g. moisture content), local weather at the time of the fire and longer-term weather conditions (Vernooij et al., 2023) as well as fire intensity (i.e. flaming vs smouldering fires; Laris et al., 2021).

Lie 264-65: Why are the other BB emissions from GFED4 while BA is from GFED5?

At the time of writing the manuscript GFED5 emissions were not published (this is still the case). We decided therefore to use the updated burned area data and dynamic emission factors (both published in 2023), but use the fuel consumption and other data from GFED4.1s.

Editorial Comments: Line 51: remove "e.g." Fig. 2a: There is no labeling of the y-axis. Please add that. Fig. 2b: Suggest changing the three different colored blue lines to "distinct" colors (e.g., RGB) for better legibility.

Done, the new figure 2 now looks like this:

Figure 2: Conceptual representation of scenarios (a) of BA in the EDS (blue) and LDS (red) along with mean BA from 2003-2020 (black). All areas under the curve are identical, i.e. total BA does not change A value of 1 on the y axis corresponds to the maximum monthly BA in the Mean scenario. (b) shows the process of shifting burned towards the EDS in a grid cell in central Zambia (-13.6° N, 29.1° E). For this grid cell only 37% of total BA is in the LDS and can be shifted to the EDS gaussian. The remainder is not shifted, but reduced by 37%. The BA distribution in this grid cell in the EDS scenario is given by the sum of the gaussian and reduced components. In this particular example, the total BA also remains consistent.